# Sin Nombre Virus and the Emergence of Other Hantaviruses: A Review of the Biology, Ecology, and Disease of a Zoonotic Pathogen

**DOI:** 10.3390/biology12111413

**Published:** 2023-11-09

**Authors:** Andrew T. Jacob, Benjamin M. Ziegler, Stefania M. Farha, Lyla R. Vivian, Cora A. Zilinski, Alexis R. Armstrong, Andrew J. Burdette, Dia C. Beachboard, Christopher C. Stobart

**Affiliations:** 1Department of Biological Sciences, Butler University, Indianapolis, IN 46208, USA; 2Department of Biology, DeSales University, Center Valley, PA 18034, USA; 3Interdisciplinary Program in Public Health, Butler University, Indianapolis, IN 46208, USA

**Keywords:** Sin Nombre virus, Andes virus, hantavirus, HPS, HCPS

## Abstract

**Simple Summary:**

In this review, we provide updates on Sin Nombre virus (SNV) and other New World hantaviruses. We discuss the history of SNV, its biology and relatedness to other emerging New World hantaviruses, its ecology, and its disease in humans. This review highlights an important emerging virus pathogen from North America.

**Abstract:**

Sin Nombre virus (SNV) is an emerging virus that was first discovered in the Four Corners region of the United States in 1993. The virus causes a disease known as Hantavirus Pulmonary Syndrome (HPS), sometimes called Hantavirus Cardiopulmonary Syndrome (HCPS), a life-threatening illness named for the predominance of infection of pulmonary endothelial cells. SNV is one of several rodent-borne hantaviruses found in the western hemisphere with the capability of causing this disease. The primary reservoir of SNV is the deer mouse (*Peromyscus maniculatus*), and the virus is transmitted primarily through aerosolized rodent excreta and secreta. Here, we review the history of SNV emergence and its virus biology and relationship to other New World hantaviruses, disease, treatment, and prevention options.

## 1. Introduction: Hantaviruses

Sin Nombre virus (SNV, formerly Muerto Canyon Virus or Four Corners Virus), is a negative-sense single-stranded RNA (-ssRNA) virus of the order *Bunyavirales* and family *Hantaviridae*. Hantaviruses are found across the globe, but they are divided into Old World (Europe, Africa, and Asia) and New World (North and South America) viruses that are associated with different diseases. Old World hantaviruses, like Hantaan virus, Puumala virus, Dobrava virus, and Seoul virus, cause Hantavirus Hemorrhagic Fever with Renal Syndrome (HFRS). Due to the availability of vaccines and therapeutics, HFRS has a case fatality rate of less than 1% [1]. New World hantaviruses, on the other hand, cause Hantavirus Pulmonary Syndrome (HPS), which has an estimated case fatality rate in the range of 30–60%. The viruses that cause HPS include SNV, Monongahela virus, New York virus, Bayou virus, and Black Creek Canal virus in North America and Andes virus (ANDV), Araraquara virus, Leguna virus, and Choclo virus in South America (Table 1). SNV is the most common cause of HPS in the United States, and ANDV is the most common cause of HPS in South America. Together, they are responsible for most HPS disease in humans [2].

The presence of New World hantaviruses was first detected in the United States in 1993, when there was an SNV outbreak in the Four Corners region. Most cases were identified in New Mexico and Arizona. Subsequently, SNV and other hantaviruses have been identified across the U.S. The geographical distribution of these hantaviruses is determined by the rodent vectors that carry each type of virus. For SNV, the carrier is the common deer mouse, *Peromyscus maniculatus*. Hantaviruses are transmitted to humans through exposure to infected rodent saliva, urine, or feces. This review discusses the emergence, biology, host range, transmission, clinical presentation, treatment, and prevention of SNV.

## 2. Emergence of Sin Nombre Virus (SNV)

In the spring of 1993, an outbreak of an enigmatic respiratory illness began in the Four Corners region of the United States; it was characterized by severe pulmonary symptoms primarily affecting young, previously healthy individuals [3]. The index case was a 19-year-old Navajo man who presented with acute respiratory distress marked by significant pulmonary edema. The patient’s fiancée had just died of a similar disease. An additional five cases arose in the region presenting with an unexplained pulmonary illness, initiating an investigation [4]. As cases accumulated, it became evident that afflicted individuals had recent close contact with rodents, prompting a shift in focus to zoonotic transmission. A breakthrough was achieved when, in the same year, a virus later identified as Sin Nombre virus (SNV) was successfully isolated from a deer mouse specimen procured from the residence of an initial case [5]. The virus was isolated by blind passaging of the virus in deer mice followed by subsequent passage and adaptation in Vero E6 cells [6]. The isolated strain, designated NRM11, was isolated from lung tissue derived from the aforementioned deer mice. The outbreak of Hantavirus Pulmonary syndrome (HPS) in 1993 prompted a nationwide call for medical professionals to report any cases exhibiting similar symptoms. In response, a spectrum of New World hantaviruses responsible for causing HPS was identified in the USA, including Bayou virus (Louisiana), Black Creek Canal virus (Florida), and New York-1 virus (New York) [3].

Since 1993, the annual reported cases of SNV in the United States have consistently ranged between 10 and 50 cases per year, accounting for approximately 850 reported cases across 39 states to date [3]. While the total number of cases remains low, the fatality rate is high, making this virus a concern for future outbreaks. Distribution of the virus is primarily due to the geographic range of the deer mice that serve as the primary carriers of SNV (Figure 1). Several factors contribute to the prevalence of SNV. First, seasonal changes appear to exert a significant influence on virus prevalence, with the majority of cases annually occurring in early spring [7]. Research using Chilean rice rats suggested a higher rate of horizontal transmission of hantavirus between rats during the spring and summer months [8]. Human activity, such as agriculture and other man-made interventions in the environment, can disrupt the normal ecological equilibrium of deer mice, leading to increased interactions between humans and rodents carrying SNV. The virus, at times, has resurfaced in unexpected locations. In 2012, a small outbreak of HPS occurred in Yosemite National Park, claiming the lives of three tourists [3]. More recently, in June 2021, a case of SNV was reported in Michigan. This was the first reported human case of SNV in Michigan [9]. The virus continues to pose a risk, even in areas where it has not been as prevalent. However, considering the geographic range of the vectors for New World hantaviruses, like SNV and ANDV, more ecological and surveillance studies are needed to better understand the dynamics of transmission of these viruses within non-human animal hosts.

## 3. Virus Biology

### 3.1. Virus Structure and Genome Organization

Hantaviruses form mostly spherical or pleiomorphic enveloped virions covered in tetrameric spikes composed of the heterodimers of the two viral glycoproteins, Gn and Gc [10]. SNV has a -ssRNA genome that comprises three RNA segments (Figure 2) [11]. The three segments are designated as small, medium, and large segments. The small (S) segment, which is 2.06 kb, codes for the nucleocapsid protein. The medium (M) segment is 3.70 kb and codes for a glycoprotein precursor (GPC), which eventually results in the two viral envelope glycoproteins, Gn and Gc, (formerly G1 and G2, respectively). Finally, the large (L) segment is 6.56 kb and encodes the viral RNA-dependent RNA polymerase (RdRp, L pro). The nucleocapsid forms trimers that then oligomerize into a helical structure with a positively charged groove for the genomic RNAs to bind [12]. The viral genome sequence of strain NMR11, obtained in 1993 from deer mouse lung tissue, was compared with the virus isolated from human tissue post mortem (strain NMH10). There were only 16 differences between the nucleotide sequences of each strain, and none of these altered the amino acid sequences. Furthermore, there was no evidence found of segment reassortment with any other hantaviruses that have been well described up to this point [6]. This lack of reassortment suggests that SNV has likely co-existed with the deer mouse population for a long time, and prior spillover events may have been occurring without reporting.

### 3.2. Virus Entry

Like other pathogenic hantaviruses, SNV enters into endothelial cells using integrins as the receptor—specifically, the β3-integrins [14]. The viral glycoproteins, Gn and Gc, are used to facilitate viral attachment to β3-integrins and subsequent fusion with the host cell membrane [15,16]. While it is not known exactly how SNV enters the host cell, Old World hantaviruses have been shown to use clatherin-coated vesicles for entry [17]. However, ANDV was shown not to use clatherin-coated vesicles, which may suggest that New World hantaviruses may also use an alternative strategy for entry. It is still thought that the New World hantaviruses enter through an endocytic vesicle and acidification of the endosome causes a conformational change in Gc that releases the fusion peptide that allows for fusion and release of the viral genomic RNA into the cytoplasm [18]. The virus then replicates in the cytoplasm of the host cell.

### 3.3. Virus Transcription

Once in the host cell cytoplasm, it has been shown that the three SNV genome segments get transcribed at different times post infection. First, the N gene is transcribed, and those transcripts can be detected starting at four hours (h) post infection (p.i.) in Vero E6 cells. GPC mRNA starts to be detected at about 32 h p.i, and, finally, L protein is detected at 48 h p.i. Transcription is thought to occur using a prime and realign strategy for at least some transcripts. Once transcripts are made, they snatch caps from host transcripts that are in the cytoplasm, possibly at cytoplasmic processing bodies [11,16,18,19]. Interestingly, although the SNV N and L mRNAs are not polyadenylated, the GPC mRNA has a poly(A) tail that is templated by eight U residues [20]. It is thought that in absence of the poly(A) tail in the N and L mRNAs, a stem-loop structure may terminate transcription.

### 3.4. Virus Particle Formation

SNV has been observed to produce both granular and filamentous inclusion bodies in the cytoplasm of infected cells, and it has demonstrated a type of budding unique to hantaviruses. Unlike Old World hantaviruses that bud at intracytoplasmic membranes, like the Golgi, SNV virions bud at the plasma membrane [21]. The virions of SNV have been described as roughly spherical, with a mean diameter of 112 nm, including a dense lipid envelope and closely positioned surface projections, averaging less than 10 nm each, and filamentous nucleocapsids [21]. In a study from 2019, two strains of SNV were studied to determine their morphologies, and the round particles were found to have a mean diameter of 90 nm. The tubular particles were found to have a mean diameter of 85 nm and an average length of 180 nm. Both strains had irregular morphologies, but there was a normal distribution of tubular and rounded particles [13].

The structure of the Tula virus virion, an apathogenic Old World hantavirus, was determined through cryoelectron microscopy [10]. It showed that the virus particles were pleomorphic in shape. The HTNV Spike G_N_ and G_C_ structure has been solved through high resolution X-ray crystallography [22]. The glycoproteins (G_N_ and G_C_) arrange into a heterodimer where the G_N_ forms the stalk and G_C_ forms the globular head. These heterodimers then form a tetrameric spike [22,23]. These spikes form a lattice that covers most of the surface, with only small areas of the membrane exposed. The spike-to-spike interactions on the membrane may be responsible for the membrane curvature of the virion [10]. The spike glycoprotein is a class II fusion protein. However, instead of having one fusion loop that makes up the target membrane insertion surface, HTNV has three loops that make up this region [22].

### 3.5. Evasion of Innate Immunity

During SNV replication, the GPC protein is co-translationally cleaved in the endoplasmic reticulum by the host signal peptidase, producing an N-terminal fragment (Gn) and a C-terminal fragment (Gc) [24]. For NY-1 virus, the Gn fragment cytoplasmic tail was shown to inhibit RIG-1- and TBK-1-directed interferon responses by blocking the formation of TBK1-TRAF3 complexes. This inhibition delays the innate immune response to allow for virus replication and spread within the host [25,26]. It has also been shown that SNV Gn degrades in the host cell by using host autophagy machinery [24]. This degradation of Gn likely counteracts the immune evasion, allowing for later induction of type I interferons.

## 4. Ecology of SNV

### 4.1. Host Range

Unlike other *Hantaviridae* family members that use arthropod vectors, hantaviruses usually infect rodents, as is seen with SNV and other HPS-causing hantaviruses in the western hemisphere (Table 1). *Sigmodontinae* is the largest group of rats and mice in the western hemisphere, and species belonging to this group are the vectors of these HPS-causing hantaviruses. It is currently unclear why HPS- and HFRS-causing viruses only exist in the Americas, but it may have something to do with a property of New World hantaviruses and their association with American sigmodontine rodents. The *Sigmodontinae* genus, which encompasses the rodent hosts for the major North American HPS-causing viruses, is *Peromyscus*, and SNV is carried specifically by *Peromyscus maniculatus*, the deer mouse. It is relevant to note that there has been speculation that most, if not all, American rodents within the *Sigmodontinae* group will eventually be found to have a characteristic hantavirus, and that many of these viruses will cause a form of HPS [2]. Many rodents other than deer mice are thought to be dead-end hosts for SNV replication, with a notable one being woodrats (*Neotoma lepida)*, which were shown to be prone to SNV infection in a 1999 study [27]. The levels of virus in the bloodstreams of these and other dead-end SNV hosts are low enough not to warrant concern for transmission to other organisms.

SNV can be found anywhere its primary reservoir, the deer mouse (*P. maniculatus*), is located. The range of this species includes almost the entire populated area of North America, with the exception of parts of the southeastern United States, Alaska, and the most northern parts of Canada (Figure 1). *P. maniculatus* is mainly found in rural areas, which is also where infection in humans is seen. Only a few cases have occurred in suburban areas. All cases have been associated with overt exposure to deer mouse urine, feces, and saliva [2,28].

Longitudinal studies of rodent populations at multiple sites in both Colorado and Montana showed that male rats had higher seropositivity and that the fluctuations in IgG antibody titers were positively correlated with population changes [29,30,31,32]. Additionally, rates of seroconversion were highest in the late summer and mid-winter or breeding season. Studies have suggested that some of the infected deer mice were long-lived and served as trans-seasonal reservoirs for the virus [31]. It remains largely unclear how climate change, changes in human population centers (particularly in the western United States and Canada), and agriculture will impact the prevalence of deer mice and other hantavirus vectors. The 1993 SNV outbreak has been attributed to the storage of grain following a robust growing season. A recent survey looking at the use of outbuildings and a higher incidence of hantaviruses suggests that active retrieval of tools or more active calving or lambing may lead to greater potential for exposure and transmission [33].

### 4.2. Transmission of SNV

SNV is transmitted among rodents, where it causes a persistent infection. A study conducted in Syrian hamsters found that SNV was not detected in the blood of the infected hamsters, but it was found in the endothelial cells in the lungs, heart, kidney, and brain [34]. This finding suggests that the heart, lung, and brown adipose tissue are the main sites of SNV replication during persistent infection, and the effect on these tissues can be seen using specific staining procedures. Rodent-to-rodent transmission occurs via contact with bodily fluids, through confrontations between animals, or during grooming events [35]. Rodents can also transmit to humans through aerosolization of virus particles in mouse excreta or secreta (e.g., urine, feces, saliva). These aerosolized particles are breathed in by humans, which allows the virus to infect the terminal bronchiole or alveolus. Soon after this initial infection, significant viremia is detectable, thereby resulting in a systemic infection of pulmonary endothelia and, less frequently, some other parts of the body. While the virus usually enters the body via aerosolized particles, bites from infected rodents can also transmit the disease, as the virus can be found in saliva [2]. Currently, there has been no confirmed human-to-human transmission of the virus.

## 5. Clinical Presentation and Symptoms of HPS

### 5.1. Clinical Presentation, Pathogenesis, and HPS Disease

New World hantaviruses, such as SNV, are associated with the onset of Hantavirus Pulmonary Syndrome (HPS) (Table 2). During SNV infections, HPS symptoms typically onset within 1–8 weeks post exposure to the virus (often through urine, droppings, or saliva), according to the Centers for Disease Control and Prevention (CDC) [3]. Prior to hospitalization, the prodromal period and initial disease are characterized by flu-like symptoms including fever, myalgia, headache, coughs, nausea, vomiting, chills, and dizziness [2,19,36]. A 1994 report of the Four Corners outbreak showed that all patients exhibited both fever and myalgia as well as an elevated respiratory rate (greater than 20 breaths per min) at the time of admission [4]. The mean duration of symptoms prior to hospitalization during the outbreak was found to be 5.4 days [4,19]. HPS illness then rapidly transitions to pulmonary edema, hypoxemia, tachycardia, and hypotension within a 24–48 h period that often coincides with acute respiratory failure and interstitial edema [3,4]. For the Four Corners outbreak, the mean duration between the time of first onset of symptoms and death was 8 days, with a case fatality rate of 76% [4].

While no correlation was found between the duration of symptoms at the time of admission and survival during the Four Corners outbreak, it remains highly recommended that patients be admitted as early as possible after the onset of symptoms for the best chances of survival and recovery [2,37]. Analysis of the survivors of the Four Corners outbreak showed that nearly all failed to exhibit hypotension, in contrast to fatal cases, which often exhibited severe hypotension (systolic blood pressures less than or equal to 85 mmHg) [4]. Clinically, the common presentation of gastrointestinal symptoms may help differentiate the acute respiratory distress syndrome associated with HPS from other viral-associated pneumonia diseases [36]. In addition, the noncardiogenic pulmonary edema and patterns of respiratory disease are distinct in their presentation, which led to the recognition of the disease as HPS [4].

Since the initial outbreak in 1993, there have been over 700 cases of HPS to date [3]. While the mortality rate remains high (38%), SNV does not always cause serious disease, and there have been reports of asymptomatic cases of HPS [3]. Patients with SNV who do not develop more serious illness show no major lingering effects, other than fatigue, myalgia, and shortness of breath. It is unclear whether these reports are a result of the viral challenge itself [2,3]. Other explanations of these reported symptoms suggest they could also be a consequence of an overactive immune response or potentially damaging therapies administered in the hospital. Some follow-up studies on survivors of SNV infection have demonstrated decreased flow in small airways, increased residual volume of the lungs, and decreased capacity for oxygen diffusion [2].

### 5.2. Detection and Diagnosis of New World Hantavirus Infections

Diagnosis of HPS and, more specifically, SNV infection, can be accomplished through a combination of immunologic assays and clinical presentation (Table 3). According to the CDC, a combination of a positive serological test result, the identification of viral antigen in histological samples, or amplifiable viral RNA in either blood or tissue, along with a compatible history of HPS, is diagnostic for HPS [3]. Enzyme-linked immunosorbent assays (ELISA) have been developed to detect IgM antibodies to SNV as well as to diagnose acute infections with other hantaviruses. An IgG test can be utilized with an IgM capture test. Both acute- and convalescent-phase sera could reflect a four-fold rise in an IgG antibody titer in the presence of IgM in acute-phase sera to be diagnostic for the hantaviral disease. Due to the longevity of IgG antibodies, the sera of patients can retain hantavirus-specific antibodies for years. From serological studies, subclinical or inapparent infections with the virus are rare [3]. Immunohistochemical testing for formalin-fixed tissues with specific monoclonal and polyclonal antibodies have also been successfully used to detect hantavirus antigens and support the diagnosis of hantaviral infections [38]. Lastly, antiserum has also been effectively used in Western Blots for antigenic recognition of New World hantaviruses [3].

## 6. Treatment and Prevention

### 6.1. Therapeutic Options for Treating HPS

Due to the low occurrence and the timeline for diagnosing hantavirus infections, treatment options are limited if not started prior to the onset of viremia (Table 4) [3,39,40]. Currently, the primary recommendation for treatment remains supportive care in an ICU with close monitoring [3]. Supplemental oxygen is recommended for hypoxia as the disease enters the cardiopulmonary phase, and care should be taken to not administer too many fluids due to the impacts of HPS on endothelial integrity and edema.

Several different therapeutic options have been explored experimentally. A study evaluating ribavirin use in a deer mouse model showed that ribavirin may be able to reduce infection in concert with human plasma from a seroconverted SNV patient [41]. Use of ribavirin in a lethal ANDV hamster model showed protection from lethality at a subtoxic concentration, and it could reduce disease even after long durations after initial infection [42]. However, clinical studies evaluating the emergency use of ribavirin for HPS during the Four Corners outbreak did not find a significant benefit in survivorship between patients with HPS and those without or on placebo [43,44]. A general conclusion related to use of ribavirin is that once the disease progresses beyond the initial prodromal period and into the cardiopulmonary phase, ribavirin is likely ineffective. Consequently, the CDC does not currently recommend use of ribavirin to treat HPS disease [3].

Other treatments explored for treating New World hantavirus infections include favipiravir, a pyrazine derivative, and vandetanib, a tyrosine-kinase inhibitor of the vascular endothelial growth factor receptor 2 (VEGF2). Favipiravir has been evaluated for efficacy against both SNV and ANDV infection models in hamsters [45]. Against SNV, favipiravir was shown to reduce RNA detection of the virus in the blood and lower antigen detection in the lungs. When used in the ANDV infection model, favipiravir protected against lethality and, consistent with SNV treatment, also effectively reduced both blood RNA and lung antigen levels. However, similar to studies with ribavirin, favipiravir was mostly ineffective when administered after the onset of viremia. Vandetanib was initially explored as a treatment for HPS due to the increased endothelial permeability and edema associated with HPS disease. In an ANDV lethality model in hamsters, pre-treatment with vandetanib was associated with delayed lethality and increased survivability [46]. Similarly to vandetanib, the steroid methylprednisolone was explored as a treatment option in HPS patients in Chile to limit the proinflammatory response to hantavirus [47]. However, a follow-up study revealed that methylprednisolone was not clinically effective for HPS [48].

### 6.2. Immunotherapy

Studies evaluating both monoclonal and polyclonal antibodies derived from SNV and ANDV infections have demonstrated protection in a hamster model [41,49,50]. These studies suggest that both natural infection and immunization are capable of inducing protective neutralizing antibodies. Recent efforts to map antigenic sites on the SNV and ANDV attachment glycoproteins have been largely successful, and current efforts are underway to develop prophylactic treatment options [15,50,51,52].

### 6.3. Hantavirus Vaccine Development

While there remain no FDA- or WHO-approved vaccines for New World hantaviruses and commercial development seems unlikely currently due to the low incidence of HPS disease, several studies are currently underway in attempt to develop vaccines against both SNV and ANDV. A recent viral-vectored vaccine employing a vesicular stomatitis virus (VSV) backbone and expressing either the SNV GPC or ANDV GPC was able to induce cross-reactive antibody responses and protect against lethal challenge in an ANDV model [53]. This study demonstrates that the immune responses of SNV and ANDV are immunologically cross-reactive. A follow-up study showed that the SNV vaccine was capable of being used as a bait-style vaccine to immunize deer mice [54]. Deer mice showed reduced viral RNA in both the blood and lungs and exhibited reduced transmission. This study illustrates that it may be possible to immunize vectors in the wild through a baiting strategy.

### 6.4. Prevention of HPS

The CDC notes on their website that people should avoid or minimize contact with rodents in the home, workplace, or while camping. Spaces for rodents to enter human habitats should be sealed and kept clean and clear of any food that might attract rodents, and traps should be placed around areas where there are signs of rodent infestations.

## 7. Discussion and Future Outlook

Globally, there continues to be an increasing incidence of emerging viral pathogens. Since the discovery of SNV in 1993, there have been several additional hantaviruses discovered across North and South America. New World hantaviruses, such as SNV and ANDV, are associated with HPS and are of considerable health concern due to their high mortality. Currently, there remain limited therapeutic options for treatment and no approved vaccines. Fortunately, cases of HPS remain low. As of the end of 2021, the CDC reported a total of 850 cases of HPS in the United States since reporting began in 1993. However, cases of HPS have been identified in 39 states as well as the District of Columbia, thus stretching from coast to coast in the United States. The prevalence of the rodent vectors for New World hantaviruses currently appears stable. Yet, it is unclear how climate change and demographic shifts, such as the continued migration of people from rural to urban settings, will impact both rodent populations and the potential for transmission to people. As evidenced by the Four Corners outbreak, favorable crop yields tend to be associated with increases in vector populations. In addition, there remains low awareness of hantaviruses or their diseases among people living within the geographic ranges of these pathogens. Lack of awareness and familiarity with the ecology of hantaviruses may inadvertently lead to conditions that might favor explosive growth in vector populations and increase the potential for transmissibility. Furthermore, due to antigenic similarities between SNV and ANDV, it remains unclear whether there remains a potential for New World hantaviruses to recombine if a co-infection event occurs. While there does not appear to be considerable overlap in the known vectors of New World hantaviruses, there is a need for continued ecological study to track changes in both vector and virus incidence.

More research is needed to better understand the biology of New World hantaviruses, the ecology of their vectors, and their pathogenesis in humans. The availability of rodent models for infections of both SNV and ANDV has proven helpful for better understanding pathogenesis within these vectors as well as evaluating the potential for both therapeutics and vaccines to mitigate disease. However, ecological studies investigating the abundance of SNV and related hantaviruses in nature and additional potential vectors for transmission would be beneficial to identify hotspots or areas with the greatest likelihood for exposure. Collectively, advances in both understanding viral antigenicity and vaccine design suggest that targeted antibody-based therapeutics and vaccines are possible, and that they may contain broadly neutralizing potential against heterologous New World hantaviruses. Recent advances in the development of both artificial intelligence and protein dynamics mapping combined with recent discoveries pertaining to the structure of hantavirus virions may permit better models for understanding viral replication, targets for therapeutic design, and vaccine development. While there remains a low likelihood of commercial development of vaccines for SNV and ANDV currently based on the low incidence of HPS cases, these studies provide promise for improved clinical outcomes for future cases of hantavirus infections.

## 8. Conclusions

SNV and other New World hantaviruses possess the potential for severe disease. Due to their transmission by different rodent vectors, it remains unclear how changes in both human demographics and climate change will impact viral abundance and exposure events in the future. While our understanding of hantaviruses has greatly expanded over the last few decades, future research is needed to better understand the biology, ecology, and pathogenesis of these viruses.

## Figures and Tables

**Figure 1 biology-12-01413-f001:**
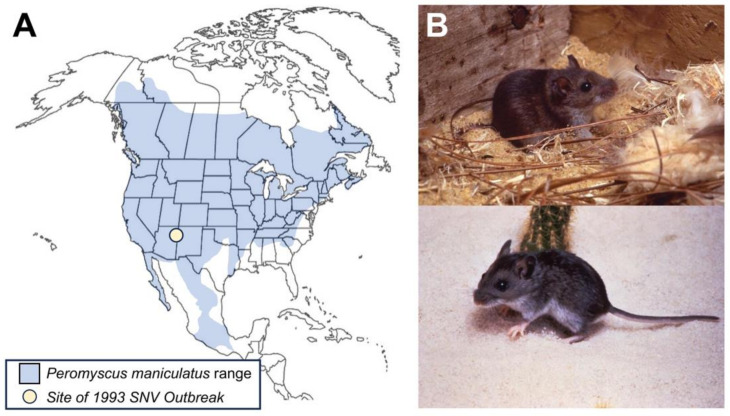
The geographic range and appearance of the deer mouse (*Peromyscus maniculatus).* (**A**) Geographic range in blue is shown for the *P. maniculatus* in North America. A yellow dot indicates the location of the Four Corners SNV outbreak from 1983. (**B**) Images of *P. maniculatus* adult mice (images taken by the CDC).

**Figure 2 biology-12-01413-f002:**
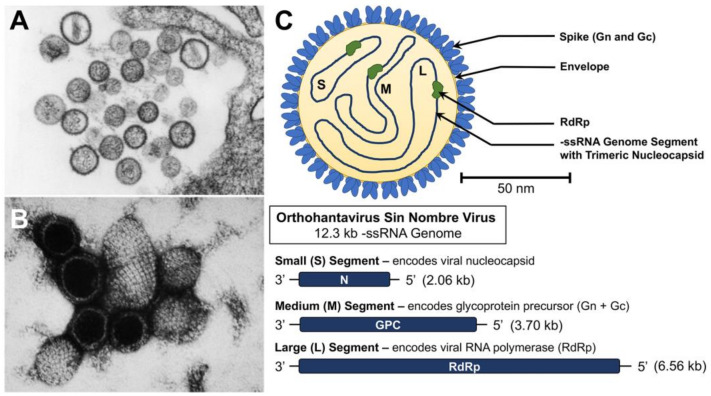
Structure and genome of Sin Nombre virus (SNV). (**A**,**B**) Transmission electron microscope (TEM) images of SNV virions (images taken by the CDC and made available through the Public Health Image Library). (**C**) Virion schematic showing the key structural and replication machinery of the virus (top) and genome organization (bottom). The sizes and gene components of the three -ssRNA genome segments of the virus are described. A scale bar has been added for the virion schematic based on reported structural analysis of SNV virions [13].

**Table 1 biology-12-01413-t001:** New World hantaviruses and their known vectors.

Virus	Vector	Location
Sin Nombre virus	*Peromyscus maniculatus*	USA
Monongahela virus	*Peromyscus maniculatus*	USA
New York virus	*Peromyscus leucopus*	USA
Bayou virus	*Oryzomys palustris*	USA
Black Creek Canal virus	*Sigmodon hispidus*	USA
Andes virus	*Oligoryzomys longicaudatus*	Argentina and Chile
Araraquara virus	*Necromys lasiurus*	Brazil
Leguna Negra virus	*Calomys laucha*	Paraguay
Choclo virus	*Oligorzomys fulvescens*	Panama

**Table 2 biology-12-01413-t002:** Clinical presentation and symptoms of HPS.

Period of Disease	Symptoms
Prodromal	Fever
	Myalgia
	Headache
	Cough
	Nausea
	Vomiting
	Chills
	Dizziness
	Elevated Respiratory Rate
Disease Onset	Pulmonary Edema
	Hypoxemia
	Tachycardia
	Hypotension
	Acute Respiratory Failure
	Interstitial Edema

**Table 3 biology-12-01413-t003:** Diagnostic options for treating HPS.

Diagnostic Method	Description
Serological Testing/ELISA	Detection of IgM antibodies specific to hantavirus; a four-fold rise in IgG antibody is considered diagnostic for hantavirus disease.
Immunohistochemical and Western Blot	Positive detection of viral antigen in histological samples; detection of hantaviral antigens by Western blot using hantavirus-specific antisera.

ELISA, Enzyme-linked immunosorbent assay; Ig, Immunoglobulin.

**Table 4 biology-12-01413-t004:** Therapeutic options for treating HPS.

Therapy	Compound Name	Virus Used	Model Used	Details
Antiviral	Favipiravir * Vandetanib	ANDV, SNV, SNV	Hamster	Pyrazine derivative Tyrosine-kinase inhibitor
Immunotherapy	Monoclonal antibodies Polyclonal antibodies	ANDV, SNV ANDV, SNV	Hamster Hamster	
Vaccine	SNV	ANDV	Deer mice	Bait style Viral-vectored vaccine

* Mostly ineffective after onset of viremia. ANDV, Andes virus; SNV, Sin Nombre virus.

## Data Availability

No new data were created for this study. However, the authors are willing to share any resources contained within upon request.

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
