# Peer review of "Sin Nombre Virus and the Emergence of Other Hantaviruses: A Review of the Biology, Ecology, and Disease of a Zoonotic Pathogen"

_biology, 2023, doi:10.3390/biology12111413_

Round 1
Reviewer 1 Report
Comments and Suggestions for Authors
The manuscript titled “Sin Nombre Virus and Emergence of Other Hantaviruses: A 2
Review of the Biology, Ecology, and Disease of a Zoonotic 3 Pathogen” by Jacob et al is an interesting and useful review of Sin Nombre Virus (SNV) and other New World hantaviruses. The manuscript is well organized and reasonably well written, but can be improved in terms of content, and by fixing small typographical errors as listed below:
MAJOR ITEMS:
If Figure 2, panels A and B are missing scale bars and it is not clear what reference they were taken from. I would advise showing more modern cryogenic electron microscopy images and/or structures, which contain proper scale bars, such as those in one of the few structural studies cited by the authors: Parvate, A.; Williams, E.P.; Taylor, M.K.; Chu, Y.-K.; Lanman, J.; Saphire, E.O.; Jonsson, C.B. Diverse Morphology and Struc- 386 tural Features of Old and New World Hantaviruses. Viruses 2019, 11, 862, doi:10.3390/v11090862.
The overview of virus structure and assembly is a bit scant and could be expanded to include several important references that were overlooked, such as
1) Huiskonen, J.T., Hepojoki, J., Laurinmaki, P., Vaheri, A., Lankinen, H., Butcher, S.J. and Grünewald, K., 2010. Electron cryotomography of Tula hantavirus suggests a unique assembly paradigm for enveloped viruses. Journal of virology, 84(10), pp.4889-4897.
2) Arragain, B., Reguera, J., Desfosses, A., Gutsche, I., Schoehn, G. and Malet, H., 2019. High resolution cryo-EM structure of the helical RNA-bound Hantaan virus nucleocapsid reveals its assembly mechanisms. Elife, 8, p.e43075.
3) Serris, A., Stass, R., Bignon, E.A., Muena, N.A., Manuguerra, J.C., Jangra, R.K., Li, S., Chandran, K., Tischler, N.D., Huiskonen, J.T. and Rey, F.A., 2020. The hantavirus surface glycoprotein lattice and its fusion control mechanism. Cell, 183(2), pp.442-456.
Please recall that, in the spirit of comprehensiveness, it is usually appropriate to include abundant references and citations in Reviews. It shows evidence of familiarity with the literature concerning the topic being reviewed. Thus, it would also be great for the authors to cite other reviews with a more narrow focus on specific aspects of hantaviruses, where the readers could get additional detailed mechanistic and structural information, such as:
4) Guardado-Calvo, P. and Rey, F.A., 2021. The surface glycoproteins of hantaviruses. Current Opinion in Virology, 50, pp.87-94.
5) Muyangwa, M., Martynova, E.V., Khaiboullina, S.F., Morzunov, S.P. and Rizvanov, A.A., 2015. Hantaviral proteins: structure, functions, and role in hantavirus infection. Frontiers in microbiology, 6, p.1326.
6) Meier, K., Thorkelsson, S.R., Quemin, E.R. and Rosenthal, M., 2021. Hantavirus replication cycle—An updated structural virology perspective. Viruses, 13(8), p.1561.
Finally, the Discussion and Future Outlook is a bit short and completely overlooks recent revolutions in structural biology and artificial intelligence, which may facilitate vaccine and therapeutic developments through intelligent drug design (unless this is considered out of scope for this review?).
MINOR (SPECIFIC) FEEDBACK:
In page 2, line 42, the sentence “Hantaviruses were first discovered in the United States in 1993” should be rephrased along the lines of something similar to “The presence of New World Hantaviruses was first detected in the United States in 1993”. Otherwise, the original sentence may be misinterpreted as hantaviruses in general being discovered for the first time (worldwide) in the United States, while the first hantaviruses (Old World hantaviruses) were actually discovered first in Korea in the 50s.
There is a small typo in page 4, line 121: the period in “SNV enters the host cell. Old world” should be replaced by a comma, as it otherwise interrupts the intended sentence.
In page 4, line 132: “4h.p.i” should probably be “four hours (h) post infection (p.i.)”, because a) numbers smaller than 10 should be spelled out according to most stylistic grammatical guidelines, 2) the “h” and “p.i.” abbreviations have not been defined and should be separate, as the first one refers to units, and the second defines a starting point; and 3) there should always be a space between a quantity and its units, so even if the authors chose to keep undefined abbreviations and use single digit numerals, it should be “4 h” and not “4h”.
Similarly, in line 133, 32h.p.i. and 48h.p.i should be 32 h p.i. and 48 h p.i., respectively (once the abbreviations have been defined in line 132).
In line 137, mRNAs is used in plural; therefore, the verb that follows should be “are” not “is”.
In line 139, the word “that” is unnecessary and should be removed.
Page 5, lines 149 and 150 contain more instances where quantities need a space between them and corresponding units (90 nm and 180 nm).
There is an extra-large space between “co-translationally” and “cleaved” in line 153 as well as between “for” and “virus” in line 158.
Page 5, line 191, “occur” should be “occurs” and in line 193 “Rodent” should be “Rodents” while in line 194 “eg.” should be “e.g.”, in italics.
Page 7, line to 50 “are not common are rare” is redundant (only one of “are rare” vs “are not common” should be kept).
Author Response
Overall Response to Reviewer 1: We have made all of the recommended changes and have responded to each below. Thank you for your suggestions.
MAJOR ITEMS:
If Figure 2, panels A and B are missing scale bars and it is not clear what reference they were taken from. I would advise showing more modern cryogenic electron microscopy images and/or structures, which contain proper scale bars, such as those in one of the few structural studies cited by the authors: Parvate, A.; Williams, E.P.; Taylor, M.K.; Chu, Y.-K.; Lanman, J.; Saphire, E.O.; Jonsson, C.B. Diverse Morphology and Struc- 386 tural Features of Old and New World Hantaviruses. Viruses 2019, 11, 862, doi:10.3390/v11090862.
Response: These images were taken from the CDC Public Health Image Library and did not have any original scale bars associated with them. We have added into the figure legend a reference to the CDC PHIL and have used the average diameter of virions from the Parvate et al. paper to provide a scale bar for the schematic of the virion in Figure 1. We were concerned about using more recent images published elsewhere due to copyright concerns, but appreciate the reviewer's suggestion.
The overview of virus structure and assembly is a bit scant and could be expanded to include several important references that were overlooked, such as
1) Huiskonen, J.T., Hepojoki, J., Laurinmaki, P., Vaheri, A., Lankinen, H., Butcher, S.J. and Grünewald, K., 2010. Electron cryotomography of Tula hantavirus suggests a unique assembly paradigm for enveloped viruses. Journal of virology, 84(10), pp.4889-4897.
2) Arragain, B., Reguera, J., Desfosses, A., Gutsche, I., Schoehn, G. and Malet, H., 2019. High resolution cryo-EM structure of the helical RNA-bound Hantaan virus nucleocapsid reveals its assembly mechanisms. Elife, 8, p.e43075.
3) Serris, A., Stass, R., Bignon, E.A., Muena, N.A., Manuguerra, J.C., Jangra, R.K., Li, S., Chandran, K., Tischler, N.D., Huiskonen, J.T. and Rey, F.A., 2020. The hantavirus surface glycoprotein lattice and its fusion control mechanism. Cell, 183(2), pp.442-456.
Response: We have added these references into the manuscript and have added additional content related to the key findings from these works in into Section 3.4 (lines 158 - 168).
Please recall that, in the spirit of comprehensiveness, it is usually appropriate to include abundant references and citations in Reviews. It shows evidence of familiarity with the literature concerning the topic being reviewed. Thus, it would also be great for the authors to cite other reviews with a more narrow focus on specific aspects of hantaviruses, where the readers could get additional detailed mechanistic and structural information, such as:
4) Guardado-Calvo, P. and Rey, F.A., 2021. The surface glycoproteins of hantaviruses. Current Opinion in Virology, 50, pp.87-94.
5) Muyangwa, M., Martynova, E.V., Khaiboullina, S.F., Morzunov, S.P. and Rizvanov, A.A., 2015. Hantaviral proteins: structure, functions, and role in hantavirus infection. Frontiers in microbiology, 6, p.1326.
6) Meier, K., Thorkelsson, S.R., Quemin, E.R. and Rosenthal, M., 2021. Hantavirus replication cycle—An updated structural virology perspective. Viruses, 13(8), p.1561.
Response: We have added these references into the manuscript throughout Section 3.
Finally, the Discussion and Future Outlook is a bit short and completely overlooks recent revolutions in structural biology and artificial intelligence, which may facilitate vaccine and therapeutic developments through intelligent drug design (unless this is considered out of scope for this review?).
Response: We appreciate this suggestion and have included a statement in the Discussion and Future Outlook section (Section 7) which discusses the importance of advances in these areas of scientific development and their applicability to developing vaccines and therapeutics for hantaviruses (see lines 391 - 397).
MINOR (SPECIFIC) FEEDBACK:
In page 2, line 42, the sentence “Hantaviruses were first discovered in the United States in 1993” should be rephrased along the lines of something similar to “The presence of New World Hantaviruses was first detected in the United States in 1993”. Otherwise, the original sentence may be misinterpreted as hantaviruses in general being discovered for the first time (worldwide) in the United States, while the first hantaviruses (Old World hantaviruses) were actually discovered first in Korea in the 50s.
There is a small typo in page 4, line 121: the period in “SNV enters the host cell. Old world” should be replaced by a comma, as it otherwise interrupts the intended sentence.
In page 4, line 132: “4h.p.i” should probably be “four hours (h) post infection (p.i.)”, because a) numbers smaller than 10 should be spelled out according to most stylistic grammatical guidelines, 2) the “h” and “p.i.” abbreviations have not been defined and should be separate, as the first one refers to units, and the second defines a starting point; and 3) there should always be a space between a quantity and its units, so even if the authors chose to keep undefined abbreviations and use single digit numerals, it should be “4 h” and not “4h”.
Similarly, in line 133, 32h.p.i. and 48h.p.i should be 32 h p.i. and 48 h p.i., respectively (once the abbreviations have been defined in line 132).
In line 137, mRNAs is used in plural; therefore, the verb that follows should be “are” not “is”.
In line 139, the word “that” is unnecessary and should be removed.
Page 5, lines 149 and 150 contain more instances where quantities need a space between them and corresponding units (90 nm and 180 nm).
There is an extra-large space between “co-translationally” and “cleaved” in line 153 as well as between “for” and “virus” in line 158.
Page 5, line 191, “occur” should be “occurs” and in line 193 “Rodent” should be “Rodents” while in line 194 “eg.” should be “e.g.”, in italics.
Page 7, line to 50 “are not common are rare” is redundant (only one of “are rare” vs “are not common” should be kept).
Response: We have incorporated all of these minor suggested changes into the manuscript.
Reviewer 2 Report
Comments and Suggestions for Authors
1. Please provide a Table to compare clinical presentation and symptoms of HPS.
2. Please provide a Table to compare the detection and diagnosis of New World Hantavirus Infections.
3. Please provide a Table to compare therapeutic options for treating HPS.
4. Please add a subsection in “Section 6. treatment and prevention” to describe the prevention of HPS.
5. Please add a subsection in “Section 6. treatment and prevention” to describe the current hantavirus vaccine development, including the reference you cited.
6. Please discuss the development of hantavirus vaccine, including the perspectives and challenges.
Comments on the Quality of English LanguageNo.
Author Response
Overall Response to Reviewer 2: We have incorporated all 3 recommended tables and have reorganized several of the sections of the manuscript as suggested. Thank you for your recommended changes and suggestions.
1. Please provide a Table to compare clinical presentation and symptoms of HPS.
Response: We have added Table 2 which has the clinical presentation of HPS. See line 253.
2. Please provide a Table to compare the detection and diagnosis of New World Hantavirus Infections.
Response: We have added Table 3 that shows the detection and diagnosis strategies. See line 294.
3. Please provide a Table to compare therapeutic options for treating HPS.
Response: We have added Table 4 to show the therapeutic options. See line 305.
4. Please add a subsection in “Section 6. treatment and prevention” to describe the prevention of HPS.
Response: We have added subsections within section 6 and section 6.4 specifically highlights prevention of HPS.
5. Please add a subsection in “Section 6. treatment and prevention” to describe the current hantavirus vaccine development, including the reference you cited.
Response: We have added subsections within section 6 and section 6.3 specifically discusses SNV vaccines.
6. Please discuss the development of hantavirus vaccine, including the perspectives and challenges.
Response: We created a specific subsection (6.3) which discusses both the development, perspectives, and challenges of SNV vaccine design. We also address several of these key challenges as well in our Discussion and Future Directions section (Section 7).
Reviewer 3 Report
Comments and Suggestions for Authors
Dear authors,
I have had the opportunity to review your manuscript titled "Sin Nombre Virus and Emergence of Other Hantaviruses: A Review of the Biology, Ecology, and Disease of a Zoonotic Pathogen." I would like to commend you for the comprehensive and well-structured review you have undertaken. Your work provides valuable insights into Sin Nombre Virus (SNV) and Hantavirus Pulmonary Syndrome (HPS), addressing various facets of the topic. The manuscript is well-structured, provides a thorough analysis of SNV and HPS, touching on various aspects, including biology, epidemiology, clinical presentation, diagnosis, treatment, and prevention. This breadth of coverage is highly commendable. The language and writing style used in the manuscript are clear and accessible, which is vital for conveying complex scientific information to a broader audience.
Your effort is commendable, and I believe your work can make a significant contribution to the scientific community.
I would like to offer some minor suggestions for further improving the manuscript:
1. Given the reference to "ecology" in the title, it's important to ensure that ecological factors, such as the role of rodents, habitat, and potential ecological drivers of hantavirus emergence, are thoroughly explored.
2. Consider including a section discussing potential future research directions or areas where further investigation is needed to advance our understanding of SNV and other hantaviruses.
Once again, I commend your efforts in preparing this manuscript, and I believe that with these suggested improvements, it has the potential to make a significant contribution to the scientific community. I look forward to seeing your work published and contributing to our collective understanding of SNV and HPS.
Regards,
The other reviewer.
Author Response
Overall Comments to Reviewer #3: Thank you for your helpful suggestions. We have addressed each of them in our revisions.
1. Given the reference to "ecology" in the title, it's important to ensure that ecological factors, such as the role of rodents, habitat, and potential ecological drivers of hantavirus emergence, are thoroughly explored.
Response: We changed the emergence and transmission section to Ecology of SNV (Section 4) and have added additional details on the host species and how other ecological factors contribute to spread of SNV. See lines 204 - 215.
2. Consider including a section discussing potential future research directions or areas where further investigation is needed to advance our understanding of SNV and other hantaviruses.
Response: In the Discussion and Future Outlook section (Section 7), we have added more details related to proposed areas of future research, including several ecological studies that are needed to better understand the proliferation and spread of hantaviruses within their vector populations. See lines 388 - 397.
Round 2
Reviewer 2 Report
Comments and Suggestions for Authors
1. The title of Table 3 should be changed into Diagnostic Options for Treating HPS.
2. Please show the full names for all abbreviations under the tables in Table 3 and 4.
Comments on the Quality of English LanguageNo
Author Response
We have addressed all of the recommended changes suggested by the reviewer and appreciate their feedback.